# IL-10 in Mast Cell-Mediated Immune Responses: Anti-Inflammatory and Proinflammatory Roles

**DOI:** 10.3390/ijms22094972

**Published:** 2021-05-07

**Authors:** Kazuki Nagata, Chiharu Nishiyama

**Affiliations:** Department of Biological Science and Technology, Faculty of Advanced Engineering, Tokyo University of Science, 6-3-1 Niijuku, Katsushika-ku, Tokyo 125-8585, Japan; 8320702@ed.tus.ac.jp

**Keywords:** mast cells, IL-10, allergy

## Abstract

Mast cells (MCs) play critical roles in Th2 immune responses, including the defense against parasitic infections and the initiation of type I allergic reactions. In addition, MCs are involved in several immune-related responses, including those in bacterial infections, autoimmune diseases, inflammatory bowel diseases, cancers, allograft rejections, and lifestyle diseases. Whereas antigen-specific IgE is a well-known activator of MCs, which express FcεRI on the cell surface, other receptors for cytokines, growth factors, pathogen-associated molecular patterns, and damage-associated molecular patterns also function as triggers of MC stimulation, resulting in the release of chemical mediators, eicosanoids, and various cytokines. In this review, we focus on the role of interleukin (IL)-10, an anti-inflammatory cytokine, in MC-mediated immune responses, in which MCs play roles not only as initiators of the immune response but also as suppressors of excessive inflammation. IL-10 exhibits diverse effects on the proliferation, differentiation, survival, and activation of MCs in vivo and in vitro. Furthermore, IL-10 derived from MCs exerts beneficial and detrimental effects on the maintenance of tissue homeostasis and in several immune-related diseases including contact hypersensitivity, auto-immune diseases, and infections. This review introduces the effects of IL-10 on various events in MCs, and the roles of MCs in IL-10-related immune responses and as a source of IL-10.

## 1. Introduction

Mast cells (MCs) were first discovered by Paul Ehrlich in 1878. Although they were found in various tissues throughout the body, particularly in blood vessels and tumors, the nature of MCs in biological processes had not been uncovered for decades [1,2]. In the 1900s, a clinical study reported that a surge of histamine was detected in the blood of patients with anaphylaxis [3], and subsequently, it was identified that MCs store a large quantity of histamine in their granules [4]. Furthermore, MCs were found to express high-affinity IgE receptors (FcεRI) on their surface and the aggregation of FcεRI, which, as a consequence of the cross-linkage of IgE antibodies on FcεRI by specific antigens, led to the activation of MCs [5]. Due to this, MCs were considered to play important roles in IgE-mediated immune responses, including the defense against parasitic infections and the initiation of type I allergic reactions. In addition to the context of parasitic infections and allergic diseases, MCs are also involved in bacterial infections, autoimmune diseases, inflammatory bowel diseases, cancers, lifestyle diseases, and allograft rejection [6,7,8,9,10,11]. In a recent study, MCs were shown not only to function as initiators of the immune response, but also to have critical roles in settling excessive inflammation by suppressing the activation of immune cells [12,13,14].

Interleukin (IL)-10 is an anti-inflammatory cytokine that plays an important role in inhibiting Th1 responses [15]. IL-10 has a high homology in its amino acid sequence with BCFR1, a protein encoded in the Epstein–Barr virus for escaping the host immune system [16]. Indeed, IL-10 suppresses the expression and/or function of a variety of inflammatory cytokines, such as IFN-γ, TNF, IL-1, and IL-6, in monocytes and T cells [17,18,19]. In addition to its anti-inflammatory role, IL-10 has also been reported to enhance immune events such as immunoglobulin production by B cells, the cytotoxicity of NK cells and CD8^+^ T cells, and the proliferation of thymocytes [20,21,22,23]. Thus, IL-10 has been considered to exhibit a duality in the immune response: promotive and suppressive.

There have been a number of previous studies investigating MCs and IL-10. Studies on the effects of IL-10 on the function of MCs demonstrated that IL-10 regulates the proliferation and production of cytokines and the differentiation of MCs [24,25,26]. Moreover, some studies focused on MCs as a source of IL-10, which plays an essential role in modulating the inflammatory response in specific pathologies [27,28,29]. The IL-10 derived from MCs has beneficial effects in the maintenance of tissue homeostasis and has detrimental effects in several immune-related diseases, including autoimmune diseases, inflammation, rejection, and infections.

This review focuses on the effect of IL-10 on MCs and on the mechanisms by which MCs control inflammatory responses as IL-10-producing cells.

## 2. The Effects of IL-10 on Mast Cells

### 2.1. IL-10 Receptor Signaling

The IL-10 receptor is composed of two subunits, a ligand-binding subunit IL-10R1 (*IL10RA*) and an accessory subunit IL-10R2 (*IL10RB*). The expression of IL-10R1 has been detected in most hematopoietic cells, and the expression of IL-10R2 has been observed extensively (reviewed in [30]). IL-10R-dependent signaling generally activates the JAK-STAT pathway, resulting in the STAT3-mediated expression of the anti-inflammatory genes. MCs are recognized as IL-10R-expressing cells since *Il10ra* cDNA was originally isolated from the mast cell line and macrophage cell line [31]. In the transcriptome data of human skin-derived MCs in FANTOM5 [32], the *IL10RA* transcript was detected at a lower level compared with other cells, whereas that of *IL10RB* was detected at an average level. Considering that MCs are highly heterogeneous depending on the origin and location in the body, detailed analyses regarding other types of MCs are required to clarify the IL-10 sensitivity of MCs.

### 2.2. Proliferation and Apoptosis

MCs are myeloid cells that develop postnatally from hematopoietic stem cells in the bone marrow [33]. Recent fate-mapping studies revealed that skin MCs are initially derived from the yolk sac and are replaced with definitive MCs in the adult [34] and that MCs originating from early erythro-myeloid progenitors, late erythro-myeloid progenitors, and definitive hematopoietic stem cells dominate in adipose tissues, connective tissues, and mucosa, respectively [35]. IL-3 and stem cell factor (SCF), which are growth factors for hematopoietic cells, play essential roles in the development of MCs in mice [36,37]. The necessity of IL-3 and SCF is supported by the fact that primary MCs and certain MC-related cell lines are generally maintained in the conditioned medium supplemented with IL-3 and/or SCF. Furthermore, other cytokines, including IL-10, are also known to be involved in MC development (Table 1).

In 1995, IL-10 was identified as a functional proliferator in MCs [38]. MCs collected from the mesenteric lymph nodes of C57BL/6 mice infected with *Trichinella spiralis* were examined using a colony formation assay to assess the proliferation activity of various cytokines such as IL-3, SCF, IL-4, and IL-10. This study revealed that the colony formation of MCs was promoted by IL-10 and was dramatically accelerated via the combination of IL-10 and IL-4. Whereas IL-4 enhanced the proliferation of MCs without affecting the morphology, IL-10 significantly increased the cell diameters and increased the granules in MCs, indicating that IL-10 promotes not only the proliferation but also the maturation of MCs [38]. In the following year, the effect of these cytokines on MCs was revealed by a study of splenic MCs [39]. The proliferation of MCs isolated from the spleen was facilitated via the treatment with IL-10 or IFN-γ, but not with IL-4. In the case of MCs from the upper airway tract, the growth of airway MCs was effectively induced by IL-4 and IL-10 at the same level as those of bone marrow-derived MCs (BMMCs), whereas airway MCs showed a lower sensitivity to SCF- and IL-3-induced proliferation in comparison with BMMCs [40]. These findings support that IL-10 generally promotes MC proliferation, and the efficacy of IL-10 as a proliferator is affected by the combination of other cytokines, depending on the microenvironmental conditions.

IL-10 has also been shown to function as an apoptosis inducer against MCs. It was reported that apoptosis in BMMCs was gradually induced during 3 days of culture in the presence of IL-4 and/or IL-10 [44]. The IL-10-induced apoptosis of MCs was accompanied by an increase in the expression of a death receptor, Fas, and a decrease in the expression of apoptosis inhibitors, Bcl-xl and Bcl-2 [44]. A further detailed study demonstrated that apoptosis induced by a combination of IL-4 and IL-10 occurs in a p53-dependent manner, which is involved in mitochondrial dysfunction with a decreased potential of membrane activity caused by Bax, an apoptotic factor of mitochondria [48]. Another mechanism has been identified, whereby IL-4 and IL-10 reduced the expression of IL-3R on MCs and downregulated STAT5 phosphorylation, implying that the inhibition of IL-3 signaling leads to the apoptosis of MCs [45]. The IL-4/IL-10-induced apoptosis of MCs was also observed in humans, because the number and viability of human MCs derived from cord blood and grown in SCF-containing medium were significantly decreased by the supplementation of IL-4 or IL-10, and they were synergistically reduced by IL-4 and IL-10 [41,46]. Interestingly, although the effect of IL-10 on the induction of cell death was similarly observed in MCs of humans and C57BL/6 mice, BMMCs of BALB/c mice were resistant to IL-10-induced cell death in association with the intensity of IL-10R expression on their cell surfaces, whereas IL-4 induced almost the same extent of apoptosis in BMMCs derived from C57BL/6 and BALB/c mice [41].

These conflicting effects (promoting proliferation and apoptosis) have been distinctly demonstrated via experiments using knockout (KO) mice. The involvement of IL-10 in the proliferation of MCs was assessed using a *T. spiralis* infection model using *Il10*^-/-^ mice of the C57BL/6 background; it was revealed that IL-10 KO resulted in a prolonged parasitic infection due to the reduction in MC proliferation and MCPT-1 production in MCs [43]. We should note that the decrease in MC proliferation in *Il10*^-/-^ mice was observed in the early stages of infection, whereas there was no significant difference in the degree of MC proliferation between wild-type (WT) and *Il10*^-/-^ in the late stage. These results suggest that IL-10 plays a critical role in the temporary expansion of MCs against pathogens but it is not necessary for the proliferation of MCs in the steady state and the late phase of inflammation. Subsequently, the growth of *Il10*^-/-^ MCs was also investigated in an in vitro experiment. The number of MCs, which was monitored as FcεRI-expressing cells during the generation of BMMCs for 3 weeks of cultivation in the presence of IL-3 and SCF, was increased more in *Il10*^-/-^ BMMCs than in WT BMMCs [43]. Under this experimental condition, IL-10 was detected in the culture supernatant of WT-BMMCs, and the IL-10 concentration increased with the days of culture along with the increase in MC population. These observations demonstrate that the IL-10 released from MCs suppresses the development of MCs in an autocrine manner. In addition to the endogenous IL-10, the exogenous IL-10 exhibited a suppressive effect on BMMC development, because the addition of recombinant IL-10 inhibited the cell survival of BMMCs [41]. However, the effect of IL-10 was the reverse in BALB/c mice. The proliferation rate of *Il10*^-/-^ BMMCs of the BALB/c background generated with IL-3 and SCF was lower than that of BALB/c-WT [42]. Moreover, the BALB/c-WT BMMCs that were transferred into BALB/c- *Il10*^-/-^ mice rapidly disappeared in six days, whereas the BALB/c-WT BMMCs that were transferred into BALB/c-WT mice were apparently detected, indicating that IL-10 has an effect as a growth factor on MCs in BALB/c mice [42].

Collectively, IL-10 functions not only as a growth factor of MCs, but also as an inducer of MC apoptosis. This dual role may contribute to a negative feedback regulation in the context of inflammation: IL-10 allows the transient expansion of MCs and then leads to the termination of inflammation by the induction of MC apoptosis. Interestingly, BALB/c mice are resistant to IL-10-induced apoptosis, which suggests that IL-10 simply accelerates the growth of MCs in some strains. The effects of IL-10 on MC proliferation and apoptosis seem to be interchangeable depending on localization, pathophysiology, and mouse strain.

### 2.3. Differentiation

MCs developed from hematopoietic stem cells in the bone marrow migrate into the peripheral blood in an immature state and differentiate according to the microenvironment in peripheral tissues [33,38]. Therefore, the MC phenotype varies greatly depending on the tissue localization, and MCs are divided into two subtypes in mice, connective tissue MC (CTMC) and mucosal MC (MMC), which can be characterized by the expression profile of specific mast cell proteases (mMCPs) [25,49,50,51]. Several cytokines have been identified to regulate the terminal differentiation from immature MCs toward MMCs or CTMCs: IL-3, TGF-β, and IL-9 induce the development of MMCs specifically expressing mMCP-1 and mMCP-2, whereas CTMCs, whose hallmarks are mMCP-4 and mMCP-5, are induced by SCF and IL-4. In humans, MCs are also classified into two types, MC_TC_ and MC_T_, in terms of the expression patterns of the proteases [52,53].

In a previous study analyzing the expression of mMCPs in BMMCs derived from BALB/c mice, it was found that IL-3-induced BMMCs expressed mMCP-4 in a steady state, and the expression of mMCP-1 was upregulated following the treatment with IL-10 [24]. Furthermore, the stimulation by IL-10 drove BMMCs to strikingly express mMCP-2, which was suppressed via the treatment with actinomycin D and cycloheximide, implying that the increase in mMCP-2 expression is induced at the transcriptional level and is mediated through the translation of other proteins, respectively [25]. A detailed study using the nuclear run-on system revealed that the transcripts of mMCP-1 and 2, which were detected even in the absence of IL-10, were remarkably upregulated by IL-10 stimulation because of the increase in transcription and mRNA stability. The mRNA expression of mMCP-2 increased 24 h after IL-10 treatment, followed by an increased protein expression from day 3 to 2 weeks later. Intriguingly, the IL-10-induced expression of mMCP-2 mRNA was no longer detected in MCs after the 5-day culture in the medium without IL-10, and mMCP-2 proteins also completely disappeared a week later [54].

Overall, IL-10 was demonstrated to increase the expression of mMCP-1 and 2, suggesting its function as a promotor of MC differentiation into MMCs. Moreover, these findings suggest that the subtype-specific protease expression of MCs exhibited plasticity and that MC differentiation may be flexibly altered depending on the tissue microenvironment.

### 2.4. Activation

MCs are localized to the skin and mucosa to obtain quick access to foreign antigens via antigen-specific IgE bound with FcεRI on the cell surface. The crosslink of FcεRI by antigen–IgE complexes induces the activation of MCs, thereby secreting inflammatory mediators such as cytokines, chemokines, and proteases. The crosslink of FcεRI is not the only way to activate MCs. Pathogen-associated molecules, damage-associated molecules, lipid mediators, and cytokines from other cells are also involved in the activation of MCs. IL-10, which is known to inhibit the production of inflammatory cytokines in various immune cells, exhibits similar suppressive effects on MC activation [26]. On the contrary, several studies have reported the positive effects of IL-10 on MC activation [42], suggesting the existence of complex mechanisms (Table 2).

The inhibitory effects of IL-10 on MC activation were reported by several studies. BMMCs treated with IL-10 for 48 h had decreased c-Kit expressions at the mRNA level in association with the lower production of TNF-α and IL-13 [47]. When BMMCs were cultured in IL-10-containing medium for 4 days, the cell surface expression of FcεRI was reduced in a STAT3 signaling-dependent manner, accompanied with the reduced transcription of α and β subunits of FcεRI, leading to the attenuation of the IgE-mediated activation of MCs [26]. As a molecular mechanism by which IL-10 inhibits MC activation, it was revealed that the protein levels of the kinases Syk, Fyn, and Akt were downregulated in IL-10-treated MCs. An in vivo experiment using mice showed that IL-10 administration in the abdomen for 4 days reduced the expression of FcεRI in peritoneal MCs and the IgE-induced secretion of TNF-α from purified peritoneal MCs [55]. Furthermore, CD68-IL-10 transgenic mice, in which IL-10 was expressed under the control of the CD68 promoter, was revealed to be resistant to passive systemic anaphylaxis (PSA) partly due to the reduced expression of FcεRI on MCs, suggesting that the constitutive stimulation by excess amounts of IL-10 improved the MC-mediated allergic responses in vivo [55]. These results support the suppressive effect of IL-10 on the antigen-dependent activation of MCs.

The suppressive effect of IL-10 on IgE-induced activation was also observed in mouse peritoneal MCs (PMCs) and human MCs [56]. IL-10 suppressed the production of inflammatory mediators from PMCs following various stimuli such as anti-IgE, LPS, PGE_1_, and PGE_2_ [56]. Interestingly, the treatment of PMCs with anti-IL-10 antibodies increased the TNF-α production, suggesting that PMCs produce IL-10 by themselves via the regulation of their activation in an autocrine manner [57]. IL-10 is also a critical regulator of antigen-dependent degranulation and the production of cytokines in human MCs. Human cord blood-derived MCs treated with IL-10-neutralizing antibodies enhanced the production of inflammatory mediators, such as leukotriene C_4_, D_4_, E_4_, TNF-α, IL-5, and IL-8, suggesting that the autocrine-regulation mechanism is effective for human MCs [58,60].

Conversely, there are studies demonstrating the enhancement of MC activation by IL-10. The implications of IL-10 for the onset of food allergies were noted in a study using *Il10*^-/-^ mice [42]. When BALB/c-WT and *Il10*^-/-^ mice were sensitized with OVA, the increase in serum IgE antibody levels was comparable between WT and *Il10*^-/-^ mice. However, *Il10*^-/-^ mice failed to develop the pathology of food allergy, and the number of activated MCs in *Il10*^-/-^ mice was significantly lower than that in WT mice. The development of food allergies in *Il10*^-/-^ mice was complemented with the transfer of WT BMMCs, suggesting that the IL-10 derived from MCs is crucial for the pathology of food allergies. The involvement of MC-driven IL-10 in the IgE-induced activation of MC was confirmed by in vitro experiments: the mRNA levels of TNF-α, IL-6, IL-13, and IL-4 in IgE-stimulated *Il10*^-/-^ BMMCs were lower than those in WT BMMCs, and the secretion of TNF-α and IL-13 proteins also decreased in *Il10*^-/-^ BMMCs. Furthermore, the supplementation of recombinant IL-10 for one week augmented the cytokine production ability of both WT and *Il10*^-/-^ BMMCs [42]. The enhancement of MC activation by IL-10 was also observed in another study using C57BL/6 mice [59]. A 24 h pretreatment with IL-10 enhanced the IgE-induced production of IL-6, IL-13, and mMCPT-1 in BMMCs, and similar results were obtained with peritoneal-derived MCs and human-skin MCs. Furthermore, IL-10 administration for 24 h before the antigen challenge exacerbated the IgE-induced PSA with the increased release of cytokines, eicosanoids, and histamines and the increase in the drop in body temperature. Based on the molecular mechanism by which IL-10 amplifies MC activation, it was suggested that IL-10 induced the expression of miR-155, which downregulated suppressor of cytokine signaling 1 (SOCS1) [59].

The dual effects of IL-10 on MCs seem to be related to the conflicting effects observed as proliferation and apoptosis. IL-10 enhanced MC activation following short-term treatment (less than 24 h), whereas long-term treatment with IL-10 is likely to lead toward the suppression of MC activation. We should note that the exacerbation of PSA due to temporary IL-10 administration and the loss of the capability to develop a food allergy in *Il10*^-/-^ mice indicate that IL-10 is a crucial factor for the immediate activation of MCs. Interestingly, by contrast, chronic IL-10 overexpression in mice relieved the drop in body temperature with PSA, suggesting that the activation of MCs is inhibited in the case of long-term exposure to IL-10. Analysis using PMCs revealed that the IL-10 produced by MCs themselves modulates MC activation in an autocrine manner [57]. In summary, IL-10 seems to contribute to the optimization of MC responses by regulating their growth and activation (Figure 1).

## 3. The Roles of IL-10 in Mast Cell-Related Immune Diseases

### 3.1. Contact Hypersensitivity

MCs play important roles in the immediate inflammation in the skin in response to various antigens and haptens [12,61]. On the contrary, the defensive functions of MCs have been reported in some skin inflammatory conditions such as contact hypersensitivity (CHS) [28,62,63,64]. CHS is dominated by Th1- and CD8^+^ T cell-mediated adaptive immune responses, and the pathogenesis consists of two distinct phases: sensitization and challenge [65]. In the sensitization phase, following the antigen presentation by Langerhans cells and dermal dendritic cells, antigen-specific T cells are primed for subsequent exposure, and, as a result, they are activated with re-exposure to the antigen in the challenge phase [66]. In addition to antigen-presenting cells and T cells, the innate immune cells, including MCs, are also involved in the induction of a sufficient CHS response (Table 3).

Controversial observations have been reported in regards to the roles of MCs in CHS. Studies using spontaneous MC-deficient mice have shown contradictions: one study reported the attenuation of CHS in *Kit**^W/W-v^* and Sl/Sld (WBB6F1/kit-Kitsl/Kitsl-d/Slc) mice [61,67], whereas another study indicated that the pathologies of CHS in *Kit**^W/W-v^* mice and *Kit^Wf/Wf^* mice are similar to that in WT mice [61,67]. To clarify the conflicting observations in the studies using spontaneous mutant mice, a study using gene-targeted mice was conducted [12]. When MCs were depleted by diphtheria toxin (DT) injections administered to *Mcpt5-Cre^+^iDTR^+^* mice, the CHS response induced by DNFB and FITC was attenuated due to the reduction in the antigen-specific response of T cells [12]. Detailed analysis revealed that the number of dendritic cells in the lymph nodes decreased because of the reduced migration ability of dendritic cells in the mutant mice, leading to an incomplete antigen presentation. In this study, the MC-specific KO of IL-10 did not affect the degree of ear swelling in CHS.

In contrast, MCs and MC-derived IL-10 may play inhibitory roles in skin inflammation in some circumstances [28]. A study using *Kit**^W/W-v^* and *Kit**^W-sh/W-sh^* mice revealed that the MC deficiency exacerbated ear swelling in CHS induced by haptens or by chronic irradiation of UVB. The reconstitution of WT BMMCs but not *Il10*^-/-^ BMMCs ameliorated the CHS, supporting the importance of IL-10 produced by MC for the suppression of skin inflammation. The protective role of MC-derived IL-10 was further supported by a study using a fluorescent imaging approach. In this study, DNFB-induced CHS deteriorated in three types of MC-deficient mice: *Kit^W-sh/W-sh^*, *Mcpt5-Cre^+^*; *DTA*, and *Cpa3-Cre^+^*; *Mcl-1^fl/fl^* mice. Furthermore, *Mcpt5-Cre^+^*; *Il10^fl/fl^* mice exhibited enhanced CHS pathologies, supporting the protective role of MC-derived IL-10 in skin inflammation [68].

In contrast to the above-mentioned effect of UVB inducing inflammation in skin, UVB irradiation is also reported to reduce MC-related allergic skin inflammation [62,69]. Some possible mechanisms by which UVB irradiation increases IL-10 production from MCs have been proposed. It is well known that UVB irradiation induces the synthesis of vitamin D_3_ in the epidermis. A study by Biggs et al. showed that 1a,25-dihydroxyvitamin D_3_, which is converted from vitamin D_3_ through enzymatic metabolism and functions as a VDR agonist, promoted IL-10 secretion from BMMCs in vitro [70]. The low-dose UVB-induced skin inflammation in *Kit**^W/W-v^* mice was ameliorated by the reconstitution of WT BMMCs, whereas the reconstitution of *VDR*^-/-^ BMMCs partially but significantly reduced the skin inflammation pathology of *Kit**^W/W-v^* mice, demonstrating that VDR signaling in MCs is involved in the suppression of UVB irradiation-induced skin inflammation. UVB irradiation followed by IL-10 production from MCs is also reported to suppress the germinal center reaction to inhibit antibody production via regulating transcription factors in follicular helper T cells [27].

MCs also contribute to CHS suppression via the modulation of the IL-10-producing Breg function [71]. IL-5 produced by MCs maintains the proliferation and IL-10-production ability of Bregs, which was required for the suppression of the inflammation in CHS [14]. This finding implies that at least MCs localized in the skin have a protective effect on CHS via Bregs.

### 3.2. Rheumatoid Arthritis

Rheumatoid Arthritis (RA) is among the autoimmune diseases in which the relationship with MCs is most well studied. As the pathophysiology is diverse depending on the individuals, the contribution of MCs also seems to vary. Many studies have shown evidence supporting MCs as an exacerbator in RA, but some reports suggest a protective effect [8,72,73].

Based on a mouse disease model using type II collagen, the serum level of antigen-specific IgE is inversely correlated with rheumatic symptoms [74]. Furthermore, several studies have supported the possibility that Th2-dominant conditions suppress the RA pathology by inhibiting Th1 responses [75]. Regarding the molecular mechanism by which MCs exhibit protective effects in RA symptoms, MCs have been found to suppress monocyte activation through the production of IL-10 and histamine [76]. In this study, alarmin IL-33 activated MCs, which induced the production of IL-10 and enhanced the expression of FcγRIIA in MCs, resulting in the increase in the release of IL-10 in a synergistic manner by the combination of IgG and IL-33. In addition, MCs suppressed TLR4-mediated monocyte activation, and MCs were found to directly interact with monocytes in the synovium [76]. These findings are consistent with the inverse correlation between IL-33 expression and other inflammatory markers in the synovium in human RA patients [75].

### 3.3. Antineutrophilic Cytoplasmic Antibody-Associated Vasculitis

The protective role of MC-derived IL-10 has been reported in Antineutrophilic Cytoplasmic Antibody (ANCA)-associated vasculitis (AAV). In patients with acute AAV, the development of the disease is correlated with the degree of MC accumulation as well as the number of spindle-shaped MCs (the form of activated MCs) at the lesion site [77]. MPO-AAV is an AAV disease in which the anti-myeloperoxidase (MPO) antibody is commonly detected in the sera [78]. A study using a mouse MPO model showed that the pathology was improved by disodium cromoglycate (DSCG), known as an MC stabilizer, whereas DSCG did not affect the development of MPO autoimmunity of *Kit**^W/W-v^* mice [79]. Given these findings, MCs have been considered as an exacerbator of MPO-AAV.

One study suggested a protective role of IL-10 from MCs in MPO-related inflammation [80]. Antigen-specific CD4^+^ T cells in the draining lymph nodes (dLNs) increased in *Kit^W-sh/W-sh^* mice in comparison with WT mice, and *Kit^W-sh/W-sh^* mice exhibited enhanced delayed-type hypersensitivity (DTH) responses following MPO injection on the footpad. Whereas the reconstitution of *Kit^W-sh/W-sh^* mice with WT BMMCs alleviated the inflammation, the *Kit^W-sh/W-sh^* mice that received *Il10*^-/-^ BMMCs exhibited deteriorated symptoms. Ex vivo experiments revealed that the IL-10 release in culture supernatants of dLN cells from *Kit^W-sh/W-sh^* mice was lower than that from WT mice and that IL-10 produced by MCs enhanced the immunosuppressive capacity of Tregs, which was consistent with the decrease in Treg cells in the dLNs in *Kit^W-sh/W-sh^* mice. Moreover, upon MPO immunization, MCs were observed to immigrate into dLNs and directly contact Tregs, suggesting that MCs affect the function of Tregs through cell–cell interaction, as well as the production of IL-10 [80].

### 3.4. Graft-Versus-Host Disease

Graft-Versus-Host Disease (GVHD) is a severe immune disease caused by the allogeneic reaction of transplanted hematopoietic cells from the donor against recipient cells. Although MCs were previously recognized as a driver of the pathology of GVHD, which was supported by a GVHD model study using *Kit^W-sh/W-sh^* mice as recipients [81] and by a study evaluating the effect of blocking the interaction between IgE and FcεRI in GVHD [82], an opposite role of MCs was also found in GVHD [83]. Briefly, C57BL/6-*Kit^W-sh/W-sh^* mice that received a transfer of allogeneic T cells exhibited severely exacerbated GVHD in comparison with C57BL/6-WT mice [83]. The lethality of GVHD in C57BL/6-*Kit^W-sh/W-sh^* mice was decreased by the reconstitution with WT BMMCs, but not by *Il10*^-/-^ BMMCs. Therefore, this study clearly demonstrated that MCs contribute to the suppression of transplant rejection as a source of IL-10.

### 3.5. Bladder Infection

The immune system in the microenvironment of some tissues is in a state of high suppression to avoid excessive inflammation, which is called immune privilege [84]. Immune privilege is an essential mechanism for protecting organs from tissue damage. In recent years, it has been reported that MCs play vital roles in the establishment of immune privilege in the bladder.

The bladder is an organ that stores excrement as urine. Uropathogenic *E. coli* (UPEC) is the most common bacteria in urinary tract infections (UTI), which sometimes spread to the kidneys [85]. Clinical studies have shown that antigen-specific antibodies were produced when bacterial infection occurs in the kidneys, whereas specific antibodies were not detected in bladder-limited infections [86]. It was considered that the adaptive immune response was suppressed in the bladder and resulted in bacteria more likely to reinfect in UTI cases, which is suggestive of tissue-specific immune privilege in the bladder [87]. A detailed analysis of UPEC infection in the bladder and kidney was performed using mouse models [29]. Mice infected with UPEC in the bladder failed to produce antigen-specific antibodies, and a longer time was needed to eliminate bacteria in comparison to kidney infections. Gene expression analysis in the case of bladder-limited infections revealed that the transcripts of immunosuppressive factors, such as *Socs3*, *Tgfb1*, and *Il10*, increased in the bladder but were not detected in the kidneys. MCs were identified as the primary source of IL-10 rather than Tregs and macrophages in bladder-infected mice. Further analysis using MC-specific *Il10*^-/-^ mice indicated that adaptive immunity was inhibited by the IL-10 produced by MCs in UTI infections. One possible mechanism is that MC-driven IL-10 suppressed the migration of dendritic cells into their lymph nodes, and, therefore, antigen presentation was limited, reducing the number of CD4^+^ T cells and germinal center B cells [29].

## 4. Conclusions

In this review, we focused on the roles of IL-10 in MC-related immune events and on MCs as a source of IL-10. Whereas IL-10 is essential for the spontaneous expansion of MCs, IL-10 also functions as an apoptosis inducer against MCs. In addition, IL-10 has dual effects on MC activation, namely acceleration and suppression. Although the rapid activation of MCs is required for immediate immune responses, including protection against parasite infections, the hyper and/or unnecessary activation of MCs risks the development of allergic diseases. Therefore, the function of IL-10 in augmenting the activation of MCs is essential for a rapid immune response, and the roles of IL-10 in enhancing apoptosis and inhibiting the excessive activation of MCs contribute to the maintenance of tissue homeostasis.

The IL-10 produced by MCs is involved in the pathophysiology of contact dermatitis, autoimmune diseases, and immune privilege in the bladder (Figure 2). The roles MCs play in various pathophysiological models have been revealed using mice deficient in SCF-Kit signaling, including Sl/Sld, *Kit**^W/W-v^*, and *Kit**^W-sh/W-sh^*. Although these mice were indispensable for analyzing the function of MCs in vivo, several studies have demonstrated the dysfunction with hematopoietic cells other than MCs in these mouse strains, suggesting a possibility of misleading conclusions in studies using these mice. In terms of MCs as a source of IL-10, we need to consider the MC subsets, because the IL-10-producing ability is different among MC types: briefly, human skin MCs produce little/no IL-10 [32,88], while human lung MCs express IL-10 upon IgE stimulation [89]. Although the molecular mechanism by which MCs produce IL-10 is largely unknown so far, several transcription factors involved in the expression of IL-10 have been uncovered in other immune cells and may also contribute to IL-10 production in MCs (Figure 3). In recent years, mice genetically modified by targeting genes encoding mMCPs have been generated to study the distinct function of specific mMCP-expressing cells, MCs, and basophils. Studies using gene-targeted mice have elucidated the physiological functions of MCs and basophils, which have been overlooked for a long time. Whereas *Mcpt5-Cre* mice and *Mcpt8-Cre* mice are powerful tools for analyzing CTMCs and basophils, respectively, it is still necessary to establish MMC-specific gene-targeted mice to analyze the role MMCs play in mucosal immunity-related diseases such as food allergies. Recently, we reported the mechanism of the MMC-specific transcriptional regulation of *Mcpt1* and *Mcpt2* genes [90], and we plan to generate knock-in mice targeting MMC-specific genes based on the background. These innovative tools will help us to further understand the function of MCs, which is expected to be applied to the treatment of allergies and other various diseases.

## Figures and Tables

**Figure 1 ijms-22-04972-f001:**
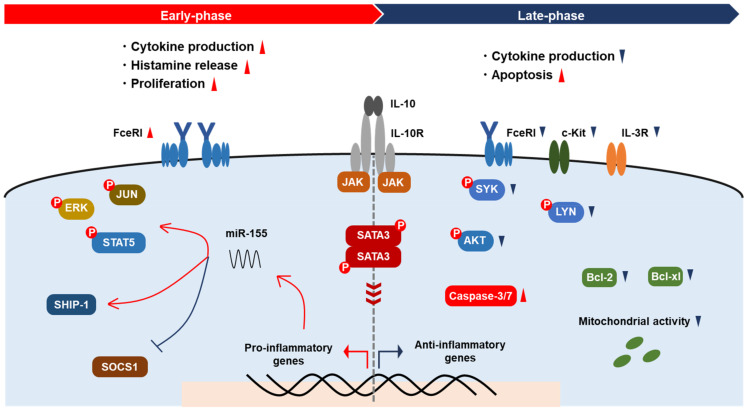
Modulation of MC activation by IL-10. IL-10 promotes cytokine production, histamine release, and the proliferation of MCs via augmentation of FcεRI signaling and miR-155 expression in the early phase, resulting in the activation of kinases and transcription factors and the suppression of SOCS1. In the late phase, IL-10 reduces cytokine production and induces apoptosis. The expression of IL-3R and c-Kit on MCs decreases via IL-10. IL-10 downregulates signaling molecules, including Syk, Lyn, and Akt, and apoptosis inhibitors Bcl-xl and Bcl-2, resulting in mitochondrial dysfunction.

**Figure 2 ijms-22-04972-f002:**
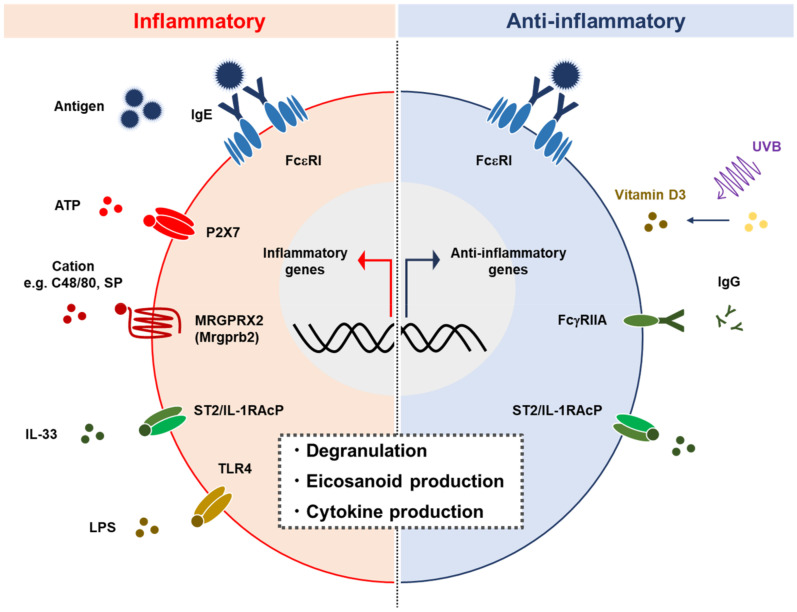
MCs play proinflammatory and anti-inflammatory roles. MCs are activated by various stimuli such as IgE crosslink, ATP, cations, IL-33, and LPS, via FcεRI, P2X7, MRGPRX2 (Mrgprb2), IL-33R, and TLR4, respectively. MC activation is followed by degranulation and the release of cytokines and eicosanoids. MCs also exhibit anti-inflammatory function including the production of IL-10 in certain circumstances [70,76]. SP: Substance P.

**Figure 3 ijms-22-04972-f003:**
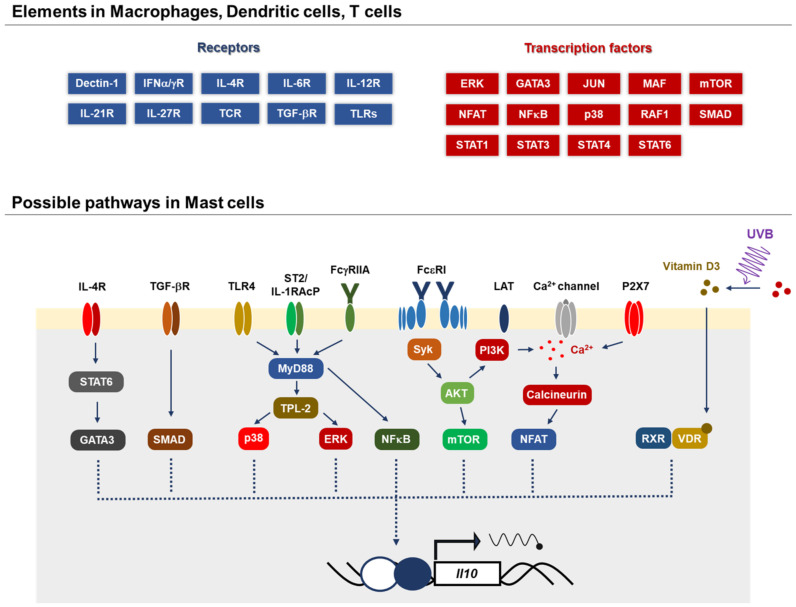
Possible mechanisms by which MCs produce IL-10. Several signal-transducing molecules and transcription factors, GATA, Smad, p38, ERK, NFkB, mTOR and NFAT, have been identified to contribute to IL-10 production in various immune cells. The molecular mechanism by which MCs produce IL-10 is largely unknown, whereas a few pathways have been reported: 1a,25-dihydroxyvitamin D_3_ and the IgE signaling-dependent pathway in the skin [70] and IL-33/ST2 and IgG/FcγRIIA in rheumatoid arthritis [76].

**Table 1 ijms-22-04972-t001:** Effects of IL-10 on proliferation and survival of MCs.

Effects	in	Strain	MC Type	Condition	Response	Ref.
Promotive	vitro	C57BL/6	Nb-infected MLN _SCF+IL-10_	7 days	Proliferation ↑	[38,39,40,41,42]
			BMMC _IL-3+SCF_	IL-10, 24 h		
			Upper airway tract _IL-3+SCF_	IL-10, 24 h		
		C3H/HeN	Splenocytes _IL-3_	IL-10, 2–4 days		
		BALB/c	BMMC _IL-3+SCF+IL-10_	8 days		
				21 days		
		IL-10^-/-^ (Balb)	BMMC _IL-3+SCF_	8 days	Proliferation ↓	
	vivo	IL-10^-/-^ (B6)	MMCs (*T. spiralis*-infected MLN)	8 days	MC number ↓	[42,43]
		NIH		Anti-IL-10 Ab i.p. 11–15 days		
		IL-10^-/-^ (Balb)	Jejunal MC	OVA-induced FA		
Suppressive	vitro	C57BL/6	BMMC _W+IL-10_	14–21 days	Apoptosis ↑	[41,44,45,46,47]
			BMMC _W_	IL-10, 14 days		
				IL-10, 6 days + IgE-XL		
		Human	Fetal liver monocytes _SCF_	IL-10, 7 days		
		C57BL/6	BMMC _W_	IL-10, 2–7 days		
		BALB/c		IL-10, 3 days		
		IL-10^-/-^ (B6)	BMMC _IL-3+SCF_	21 days	Proliferation ↑	

Subscripts in the line of MC type indicate the supplements in the culture media. Abbreviations: Balb; BALB/c, B6; C57BL/6, Nb; *Nippostrongylus brasiliensis*, MLN; Mesenteric Lymph Node, W; WHEI-3 conditioned medium, Ab; Antibody, i.p.; intraperitoneally injection, FA; Food Allergy, IgE-XL; IgE crosslink. Symbols: ↑; up-regulated, ↓; down-regulated.

**Table 2 ijms-22-04972-t002:** Effects of IL-10 on function of MCs.

Effects	Strain	MC Type	Condition		Stimuli	Response	Ref.
Suppression	C57BL/6	BMMC _IL-3_ or BMMC _W_	IL-10	4 days	-	FcεRI ↓	[26,55]
	BALB/c	BMMC _IL-3_					
	C3H						
	129						
	C57BL/6	BMMC _IL-3_		5–14 days	- or IgE		
	Human	Skin MC or Lung MC		4 days	-		
	C57BL/6	BMMC _IL-3 or W_		4 days	IgE-XL	TNF-α ↓	[26,42,47,55,56,57]
		BMMC _IL-3+SCF_		16 h			
		BMMC _W_		24 h	SCF	TNF-α, IL-13 ↓	
	Brown North rat	PMC		1–18 h	IgE-XL, LPS, PGE_1, 2_	Histamine, IL-6 ↓	
	Lewis rat				LPS	TNF-α, IL-6 ↓	
	Sprague-Dawley rat	PMC (Nb-infected)		24 h	IgE-XL	TNF-α, Histamine, Nitrite ↓	
	Human	Skin MC		4 days		Degranulation, GM-CSF ↓	
	BALB/c	BMMC _IL-3+SCF_		24 h		TNF-α, IL-4↓	
	Sprague-Dawley rat	PMC (Nb-infected)	Anti-IL-10 Ab 24 h	6 h	- or IgE-XL	TNF-α ↑	[57]
	Human	CBMC		24 h	IgE-XL	LTC_4_, D_4_, E_4_ IL-5, IL-8 ↑	[58]
	Human	CBMC		12 h, IgE-XL		TNF-α ↑	
	CD68-IL-10 Tg (B6)	-	PSA		-	TNF-α, MIP-1α ↓, Δ of body temperature ↓	[55]
Promotion	C57BL/6	BMMC _IL-3 or W_	IL-10	4 d	IgE-XL	Degranulation ↑	[26,40,42,59]
	BALB/c	BMMC _IL-3+SCF_		7 d		TNF-α, IL-6, IL-13 ↑	
	C57BL/6	BMMC _IL-3+SCF_		24 h		IL-6, IL-13, MCPT-1 ↑	
	C57BL/6	PMC _IL-3+SCF_				IL-6, IL-13 ↑	
	C57BL/6	Upper airway tract _IL-3+SCF_				Degranulation ↑	
	Human	Skin MC				MCP-1 ↑	
	C57BL/6	-	PSA (IL-10 i.p.)		-	Histamine, IL-6, MIP-1α ↑, Δ of body temperature ↑	
	BALB/c	BMMC _IL-3+SCF_	IL-10	24 h	LgE-XL	IL-6, IL-13 ↑	[42]
	IL-10^-/-^ (Balb)	-	OVA-induced FA		-	Diarrhea, MC number, Activation ↓	

Subscripts in the line of MC type indicate the supplements in the culture media. Abbreviations: Tg; Transgenic, B6; C57BL/6, Balb; BALB/c, W; WHEI-3 conditioned medium, Nb; *Nippostrongylus brasiliensis*, Ab; Antibody, PSA; Passive Systemic Anaphylaxis, i.p.; intraperitoneally injection, FA; Food Allergy, IgE-XL; IgE crosslink. Symbols: ↑; up-regulated, ↓; down-regulated, Δ; decrease.

**Table 3 ijms-22-04972-t003:** Roles of MCs and/or IL-10 in skin inflammation.

Model	Strain	Treatment	Symptom	Ref.
Picryl chloride	WBB6F-*Kit^W/W-v^*	None	Attenuated	[61]
	WCB6F-SI/SI^d^			
Oxazolone	WBB6F-*Kit^W/W-v^*		No significance	[67]
Picryl chloride	*Kit^Wf/Wf^*			
DNFB	MCPT-5-Cre DTA		Attenuated	[12]
FITC				
Urushiol	B6-*Kit^W-sh/W-sh^*		Exacerbated	[14,28,68]
DNFB	WBB6F-*Kit^W/W-v^*			
Oxazolone	B6-*Kit^W-sh/W-sh^*			
DNFB	*Kit^W-sh/W-sh^*			
	Mcpt5-Cre DTA			
	Cpa3-Cre Mcl-1^fl/fl^			
	Mcpt5-Cre *Il10^fl/fl^*			
chronic low-dose UVB	WBB6F-*Kit^W/W-v^*			
DNFB	B6-*Kit^W-sh/W-sh^*	BMMC reconstitution	Attenuated	[14,28,68]
Oxazolone				
chronic low-dose UVB	WBB6F-*Kit^W/W-v^*			
Oxazolone	C3H/HeN	pre-IgE-XL	Attenuated	[62]
	WCB6F-SI/SI^d^	pre-UVB irradiation	Loss of UVB-induced imunosuppression	[27,62]
DNP-KLH	B6-*Kit^W-sh/W-sh^*			
		BMMC reconstitution and pre-UVB irradiation		

Abbreviations: B6; C57BL/6, IgE-XL; IgE crosslink.

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
