# Peer review of "IL-10 in Mast Cell-Mediated Immune Responses: Anti-Inflammatory and Proinflammatory Roles"

_ijms, 2021, doi:10.3390/ijms22094972_

Round 1

Reviewer 1 Report

The paper by Nagata & Nishiyama summarizes the yin-yang nature in the relationship between IL-10 and mast cells. The authors give a solid overview of the different effects IL-10 exerts on MCs (pro- versus anti-proliferative, pro-/anti-apoptotic etc), also taking into account some variable results in different mouse strains (to add to the compexity). In most parts, the paper is informative but there is room for further improvement especially in terms of how information is organized  (tables are missing). A better presentation will make it a real compendium for the interested reader.

Major:

  1. A review on IL-10 should contain a paragraph (+sketch) about what is known about the IL-10 receptor and its downstream signaling components (not limited to MCs). Also, what are the major cells expressing IL10RA (and RB)?

For example, how much IL-10R expression is found in MCs relative to other cells (consult Consortia ImmGen/PMID: 27135604; FANTOM5/ PMID: 24671954)?

  1. Line 64/65:

Mast cells do not necessarily develop from bone marrow hematopoietic stem cells but are also seeded early on from yolk sac progenitors (Li et al., Immunity, PMID: 30332630

; Gentek et al., Immunity, PMID: 29858009); further recent reports suggest that and there is a strong connection between MC and erythroid differentiation.

This state-of-the-art information needs to be added to the Ms.

  1. MCs are highly heterogenous and plastic. Results for one MC type will not tell much about how another subset will behave. Therefore, it is crucial to have a clear table summarizing the different results for distinct MC subsets subdivided by in vitro generated MCs (what was the starting material if human? E.g. BM, PB, CB, iPS?) vs in vivo matured human MCs (type of tissue, e.g. lung, gut, skin, heart etc) versus mouse (specify if pMCs or BMMCs, add strain) or other species.

  1. A complementary table should give an overview of which MC subsets produce IL-10 under what conditions, i.e. murine MCs in vitro/in vivo, stratified by conditions and tissues compared to human MC subsets. Note that in humans, skin MCs produce little/no IL-10 (PMID: 14634065), while lung MCs do (PMID: 10520066)

  1. Fig 1: suggest specifying the microRNA (like the other participants), also use consistent colors, e.g. red triangles for “up” and blue triangles for “down” not the way it is now (everything blue on the right no matter if up or down).

  1. Fig 2

A bit sketchy. 1. Of the receptors given on the left only FceRI elicits degranulation, while IL-33R and TLRs do not, at least not directly. Another important receptor capable of degranulating MCs, is missing, namely MRGPRX2/Mrgprb2. The differential ability of different receptors to degranulate MCs (or not) should be visible from the fig. or at least clearly explained in the legend. Some examples of inflammatory cytokines should also be given.

  1. anti-inflammatory: the way it is now, looks like VDR is absolutely essential for IL-10 transcription and even the only TF in this scenario. It would be important to mention the TFs that organize IL-10 production IN GENERAL (not necessarily in MCs) and describe the relevant promoter/enhancer elements (at least briefly, what is known so far?). If no info is available for MCs (apart from VDR), this information can be either placed in the fig itself and marked with a “?” or at least described in the legend. Also make reference to this in the Ms proper.

Minor

  1. There are several typos/misspellings, including but not limited to the listed ones; please check entire Ms carefully

Line 28: tissues in throughout (erase "in")

Line 32: in THEIR granules

Line 193: in the abdominal -> abdomen

Fig. 1: Cytokines production -> Cytokine production

Fig. 2: in the certain -> in certain

  1. A 324 study using a mouse MPO model showed that the pathology was improved by disodium 325 chromiglycate (DSCG), known as a MC stabilizer, and that KitW/W-v mice developed more 326 severe pathology compared with WT mice [74].

-> Disease improved by DSCG but exacerbated by MC-deficiency, please explain this obvious contradiction

Author Response

First of all, we would like to thank the reviewers for their constructive comments on our manuscript. In this revised manuscript, we have attempted to address all concerns raised by the reviewers. Please see our point-by-point reply as an attached file.

Reviewer 2 Report

This review introduces the effects of IL-10 on various events in MCs and the roles of MCs in IL-10-related immune responses and as a source
of IL-10. It is a very interesting topic on the dual role of IL10, both stimulating and inhibiting  immune systems.

The review is written  concise and attractive to read. Several  aspects of MC functionality related to differentiation and activation are discussed.

IL10 significantly increased cell diameters and increased the granules in MCs, indicating that IL-10 promotes not only the proliferation but also maturation of MCs and on the other hand has the capacity to induce apoptosis. The role of IL10 in stimulating proliferation of MC seems most important in early stages of infections. A negative FB loop of IL10 produced by MC regulates the immune response. The plasticity of MC in relation to their environmental cytokines is shown and contributes to the diversity of MC. The dual role of IL10 on activation of MC was discussed in relation to FA and other immune diseases.

The figure is very elucidative and the review would improve with  a table listing the most important REFs in relation to MC and the different immune diseases. This table would certainly complement this review.

Author Response

(The authors gave the same response as above.)

Round 2

Reviewer 1 Report

The review has been greatly improved and addition of new tables and figures is highly appreciated